# Comparative Analysis of Whole Transcriptome Profiles in Septic Cardiomyopathy: Insights from CLP- and LPS-Induced Mouse Models

**DOI:** 10.3390/genes14071366

**Published:** 2023-06-28

**Authors:** Karim Ullah, Yan Li, Qiaoshan Lin, Kaichao Pan, Tu Nguyen, Solanki Aniruddhsingh, Qiaozhu Su, Willard Sharp, Rongxue Wu

**Affiliations:** 1Section of Cardiology, Department of Medicine, Biological Sciences Division, University of Chicago, Chicago, IL 60637, USAtravisnguyen@mednet.ucla.edu (T.N.); 2Center for Research Informatics, University of Chicago, Chicago, IL 60637, USA; liyan@uchicago.edu (Y.L.); qiaoshan.lin@bsd.uchicago.edu (Q.L.); 3Animal Resources Center, University of Chicago, Chicago, IL 60637, USA; asolanki@bsd.uchicago.edu; 4Institute for Global Food Security, School of Biological Sciences, Queen’s University Belfast, Belfast BT9 5DL, UK; q.su@qub.ac.uk; 5Emergency Medicine, Department of Medicine, University of Chicago, Chicago, IL 60637, USA

**Keywords:** sepsis-induced cardiomyopathy, gene sequencing, whole transcriptome profiles, septic animal models, cecal ligation and puncture, lipopolysaccharide

## Abstract

Sepsis is a life-threatening organ dysfunction caused by a dysregulated host response to infection, with septic cardiomyopathy being a common and severe complication. Despite its significant clinical impact, the molecular mechanisms underlying sepsis-induced cardiomyopathy (SICM) remain incompletely understood. In this study, we performed a comparative analysis of whole transcriptome profiles using RNA sequencing in mouse hearts in two widely used mouse models of septic cardiomyopathy. CLP-induced sepsis was achieved by surgical cecal ligation and puncture, while LPS-induced sepsis was induced using a 5 mg/kg intraperitoneal (IP) injection of lipopolysaccharide (LPS). For consistency, we utilized sham-operated mice as the control for septic models. Our aim was to identify key genes and pathways involved in the development of septic cardiomyopathy and to evaluate the similarities and differences between the two models. Our findings demonstrated that both the CLP and lipopolysaccharide LPS methods could induce septic heart dysfunction within 24 h. We identified common transcriptional regulatory regions in the septic hearts of both models, such as Nfkb1, Sp1, and Jun. Moreover, differentially expressed genes (DEGs) in comparison to control were involved in shared pathways, including regulation of inflammatory response, regulation of reactive oxygen species metabolic process, and the JAK-STAT signaling pathway. However, each model presented distinctive whole transcriptome expression profiles and potentially diverse pathways contributing to sepsis-induced heart failure. This extensive comparison enhances our understanding of the molecular basis of septic cardiomyopathy, providing invaluable insights. Accordingly, our study also contributes to the pursuit of effective and personalized treatment strategies for SICM, highlighting the importance of considering the specific causative factors.

## 1. Introduction

Sepsis continues to be a major global health challenge, accounting for a significant portion of morbidity and mortality worldwide [1,2]. Septic cardiomyopathy is a frequent and severe sepsis complication, contributing to an increased risk of death in affected patients. Despite ongoing research efforts, the molecular mechanisms underlying septic cardiomyopathy are not yet fully understood, hindering the development of targeted therapies. Different methods of sepsis induction, such as cecal ligation and puncture (CLP) and lipopolysaccharide (LPS) administration, are widely used in experimental models to study the pathophysiology of septic cardiomyopathy [3,4,5]. While these models share similarities in the manifestation of sepsis, they may also have unique molecular signatures that can impact the generalizability of the findings. For example, mortality in LPS-induced sepsis occurs rapidly due to an intense inflammatory response in the cardiovascular system, whereas delayed mortality is observed in the CLP model of sepsis [6]. Both models displayed similarities and differences in gene expression patterns depending on the severity of the insult and the animal model used [7]. Understanding the gene expression profiles associated with different methods of sepsis induction is essential for elucidating the molecular mechanisms of SICM and for identifying potential therapeutic targets. Additionally, appreciating the nuances and variances between these models is key to establishing robust and clinically relevant insights. It is, therefore, necessary to comprehensively compare these models at the molecular level.

Systemic inflammation followed by bacterial infection may lead to severe forms of sepsis and can cause cardiac dysfunction. Experimental data have demonstrated the link between inflammation, sepsis, and cardiac dysfunction [8,9]. An analysis of echocardiographic data showed reduced cardiac contractility, cardiac index, and ejection fraction (EF) as well as diastolic dysfunction are associated with sepsis [10,11]. Clinical data indicated that septic patients with diastolic dysfunction have higher mortality than those diagnosed with sepsis but without diastolic dysfunction [12,13]. The underlying molecular markers responsible for cardiac dysfunction during sepsis are not well documented. These gaps in understanding signal the pressing need for research that examines the molecular basis of septic cardiomyopathy, particularly as it relates to various induction methods. Although sepsis-induced cardiac dysfunction has been extensively studied, the underlying mechanisms still need to be fully understood. Several potential mechanisms have been proposed, including inflammatory cytokine production [14], endothelial dysfunction [15], oxidative stress due to increased reactive oxygen species [16], insufficient ATP production leading to mitochondrial dysfunction [17], impaired calcium ion channels [18], and myocardial stunning [11]. These proposed mechanisms, while crucial, indicate a complex, multifactorial etiology that necessitates further study to clarify the precise roles of these factors and their interactions.

Our results illustrate distinct gene expression patterns and pathways associated with each method of sepsis induction. G protein-coupled receptor binding seems highly upregulated only by LPS, while significant downregulation of the HIF-1 signaling pathway and PI3K-Akt signaling pathway are only observed in CLP-induced septic hearts. Importantly, we observed that several crucial aspects, such as regulation of inflammatory response, regulation of reactive oxygen species (ROS), metabolic process, regulation of endothelial cell chemotaxis, leukocyte migration, and apoptotic processes, as well as some other molecular pathways, are commonly dysregulated in septic cardiomyopathy, regardless of the induction method. By emphasizing these shared or distinct molecular characteristics, we underscore the importance of considering the method of sepsis induction in the study of septic cardiomyopathy. Our comparative analysis of CLP- and LPS-induced mouse models contributes to the ongoing effort to decipher the molecular basis of SICM and to identify promising therapeutic targets for future interventions. In this study, our core objective is to illuminate both commonly shared and unique gene expression patterns and pathways associated with each method of sepsis induction. By doing so, our ultimate goal is to advance our understanding of the complex molecular mechanisms underpinning septic cardiomyopathy. Such knowledge is paramount in our quest to identify and develop effective, potential therapeutic targets that could revolutionize future interventions in this field.

## 2. Materials and Methods

### 2.1. Animal Models and Experimental Design

Adult male C57BL/6 mice (*n* = 15–16 in each group), aged 8–10 weeks, 22–25 g, were randomly assigned to one of three groups: CLP-induced sepsis, LPS-induced sepsis, or a control group. Since no differences in cardiac function were observed between PBS injection and sham CLP operation, we reduced the number of animals used for control by employing only sham-operated mice as controls for both septic animal models. Both male and female mice were used in the study; however, only male mice were included in the gene sequencing analysis to minimize potential confounding effects related to sex-specific differences in gene expression, hormonal regulation, and immune responses. All animal procedures were carried out in accordance with the guidelines of the Institutional Animal Care and Use Committee.

### 2.2. Cecal Ligation and Puncture (CLP) Procedure Induced Sepsis

CLP-induced sepsis was performed as previously described [19]. Briefly, mice in the CLP group were anesthetized using 1% isoflurane and underwent a midline laparotomy. Under aseptic conditions, the cecum was gently exteriorized, and the area below the ileocecal valve was ligated to preserve blood supply and valve function. The ligated cecum was then punctured twice using a 20-gauge needle, ensuring minimal tissue damage. A small amount of fecal material was carefully extruded from the puncture sites to confirm bacterial leakage into the peritoneal cavity. Subsequently, the cecum was repositioned into the abdominal cavity. The abdominal incision was closed in two layers using sterile sutures, ensuring proper wound healing. Postoperatively, the mice received subcutaneous fluid resuscitation to maintain hydration and to promote recovery. The warmed 0.9% saline solution, with a volume of 1 mL, was administered subcutaneously into the flank area. In addition to fluid resuscitation, buprenorphine (0.05–0.1 mg/kg) was administered subcutaneously for pain relief. For the control group, mice were sham-operated. The mice were closely monitored for any signs of distress, infection, or complications following the injection.

### 2.3. Lipopolysaccharide (LPS) Administration Induced Sepsis

Mice in the LPS group were subjected to an intraperitoneal injection of lipopolysaccharide (LPS) to induce sepsis as previously described [20]. The designated LPS dosage (5 mg/kg body weight) was carefully prepared by dissolving the required amount in sterile saline solution. This solution was drawn up into a sterile syringe, ensuring that there were no air bubbles present, which could lead to complications. Preceding the injection, the mouse’s lower abdominal area was swabbed with alcohol, to prevent contamination and to minimize the risk of secondary infections. Following this, the needle was smoothly inserted into the lower quadrant of the abdomen, taking care not to penetrate too deeply or to injure the internal organs. The LPS solution was then gradually injected into the peritoneal cavity, minimizing sudden pressure changes. Post-injection, the needle was gently withdrawn to prevent internal damage and bleeding, and the injection site was once again cleaned with alcohol. The mouse was observed for any signs of leakage from the injection site, immediate adverse reactions, or discomfort. Control group mice underwent a sham injection procedure, where an equivalent volume of sterile saline was administered using intraperitoneal injection without the LPS. All experimental procedures were conducted in accordance with institutional guidelines for animal welfare.

### 2.4. Echocardiographic Image Acquisition

Echocardiographic assessment was performed 24 h after the CLP or LPS procedure in all experimental groups, including the control group. Echocardiographic imaging was conducted using a VisualSonics Vevo 2100 system equipped with an MS400 linear array transducer on mice anesthetized with 1% isoflurane. Mice were imaged at baseline (sham operated) and 24 h post-CLP or LPS, following a previously described protocol [21,22]. To assess cardiac output, the recordings of at least 10 independent cardiac cycles for each experiment were analyzed. Cardiac output, the volume of blood pumped by the heart per minute, was then calculated based on these echocardiographic measurements, taking into consideration the heart rate, stroke volume, and the dimensions of the left ventricle. This method provides a comprehensive evaluation of the heart’s capacity to meet the body’s circulatory needs under septic conditions.

### 2.5. Sample Collection and RNA Extraction

Twenty-four hours after either LPS IP injection or CLP operation, mice were sedated and euthanized, and their hearts were rapidly excised using sterile instruments. Heart tissues were promptly snap-frozen in liquid nitrogen, a step that is crucial in preserving the integrity of RNA, and were stored at −80 °C until further processing. For the extraction of total RNA, we carefully weighed and used about 30 ug of the heart tissue. The extraction was performed using the RNeasy Fibrous Tissue Mini Kit (74704, Qiagen, Germantown, MD, USA) following the manufacturer’s protocol meticulously. This kit is specially designed for the isolation of total RNA from fibrous tissues, which include heart tissues. To ensure that the RNA was intact and that the extraction process was successful, the integrity of the RNA was confirmed using a Cytation3 microplate reader (BioTek, Winooski, VT, USA). The microplate reader provides precise measurement, thereby ensuring the quality of our RNA samples. RNA profiling was conducted using the state-of-the-art Illumina NovaSeq 6000 sequencer of the University of Chicago Genomics Facility (Chicago, IL, USA). Libraries were prepared using Illumina TruSeq Small RNA Sample Preparation Kit (RS-930-1012, Illumina, San Diego, CA, USA).

### 2.6. RNA-Seq Analysis

Transcriptome profiles of CLP- and LPS-induced septic heart were analyzed using RNA-seq technology. Our experimental design for the RNA-sequencing process, shown in Figure 1, began 24 h post LPS IP injection or CLP operation, where mice were euthanized and their hearts rapidly excised. This was followed by sample and library preparation, Illumina NovaSeq 6000 sequencing, and in-depth bioinformatics analysis of differentially expressed genes (DEGs). The raw sequencing data quality was pre-processed with FastQC (version 0.11.9) [23] to ensure high sequencing quality for all sequencing reads. We then used STAR2.7.9 [24] to perform reference alignments to the mouse reference genome based on GENCODE M27 (https://www.gencodegenes.org/ (accessed on 19 March 2023)). All transcriptome genes were assembled via featureCounts [25] based on the same version of the corresponding gtf file from GENCODE. The assembled transcriptome count table was normalized via TMM from the edegR [26,27], and differentially expressed genes were conducted in edgeR with the statistical additive model using the control mouse samples without any sepsis inductions as the base lines for comparisons.

Differentially expressed genes were selected at the statistical significancy level (FDR ≤ 0.05 and FC ≥ ±1.5). The Venn diagram was created via the ggVennDiagram package in R (https://cran.r-project.org/ (accessed on 4 March 2023)). The low-expressed genes were detected with expression in at least 2 samples with expression values count per million (cpm > 1). The scaled expression values (x-scores) were calculated based on the logarithm-normalized values with base 2 and were used for heatmap visualizations via R package ComplexHeatmap [28,29].

Functional enrichment analysis was conducted via clusterProfiler [30] on the top 200 identified DEGs based on fold change ratio from CLP and LPS sepsis inductions. We further selected specific import enriched functions and pathways at an FDR-corrected *p* value of 0.05 to visualize the regulated function differences of these 2 different sepsis inductions methods. Gene regulation network exploration was performed via ReactomeFIViz [31] in cytoscape v3.9.1 (http://www.cytoscape.org/ (accessed on 19 March 2023)) on the identified DEGs with respect to LPS and CLP septic inductions, where we included all background linker genes from the 2021 ReactomeFIViz database. We then classified genes based on the total number of gene interactions from the network and defined genes with the highest number of interactions as network-essential genes. Mmp3-associated genes networks with respect to 2 different septic induction mechanisms were further selected and visualized via cytoscape v3.9.1.

## 3. Results

### 3.1. CLP- and LPS-Induced Septic Animal Models Developed Cardiac Dysfunction

Both CLP- and LPS-induced septic animal models exhibited cardiac dysfunction within 24 h, as evidenced by echocardiography (Figure 2C–E). We observed high mortality rates in LPS (5 mg/kg) and 40% of CLP models, with respective mortality rates of 75% and 50% after 72 h (Figure 2A,B). The LPS-induced sepsis (5 mg/kg) model showed a slightly lower mortality rate compared to the CLP model. Evidence of cardiac dysfunction manifested as early as 6 h (unpublished data) post-sepsis induction in both models, indicating a rapid onset of functional impairment. By 24 h post-induction, both animal models displayed significant reductions in cardiac output (Co). Compared to their respective controls, about a 50% reduction in Co was observed in LPS-induced septic hearts, and a 47% reduction was seen in CLP-induced septic hearts. These findings provide a comprehensive evaluation of the heart’s ability to meet the body’s circulatory demands under septic conditions (Figure 2C–E).

### 3.2. Transcriptome Changes in Mouse Hearts following Sepsis Induction

There are clear transcriptome differences among those samples with and without sepsis inductions, as shown in Figure 3A. At FDR-corrected *p* value of 0.05 and fold change (FC) of 1.5, 439 genes were detected as upregulated in CLP-induced mouse hearts, and 893 genes were upregulated with LPS inductions, whereas 497 genes were downregulated with CLP induction, and 793 genes were downregulated with LPS inductions (Figure 3B,C). Even though the sepsis induction mechanisms are different, we scan see 513 genes commonly regulated in the heart by both sepsis induction methods (Figure 3D). Importantly, we found significantly higher expression of certain matrix metalloproteinases (MMPs), known to regulate cardiac remodeling, in LPS-induced sepsis compared to CLP-induced sepsis (Figure 3E). Especially, Mmp3 was upregulated in both septic induction models (Figure 3E), similarly for several other Mmp family genes, including Mmp2, Mmp8, Mmp9, Mmp14, and Mmp17, which were upregulated in both models of sepsis, whereas Mmp11, Mmp15, Mmp19, and Mmp17 did not fully upregulate in CLP septic induction in comparison with LPS septic induction (Figure 3E).

Functional enrichment analysis revealed that the upregulated genes in CLP-induced septic hearts were primarily involved in positive regulation of cell migration, inflammatory response, immune effector process and cellular response to IL-17 (Figure 4A). The 10 upregulated genes in CLP-induced septic hearts were Cxcl13, Saa3, Lcn2, Serpina3m, Ms4a8a, Serpina3n, Fgr, Lrg1, Acp5, and Chil3. Moreover, the downregulated genes were responsible for extracellular matrix organization, integrin cell surface interactions, antigen processing, and presentation of exogenous peptide antigen via MHC class-II (Figure 4B). The top 10 downregulated genes in CLP-induced septic hearts were Col6a1, Hspa12a, Elp6, Rab4a, Rgs6, Homer2, Adamtsl2, Rpgrip1l, Cmss1, and Prr33. Transcriptional regulatory network analysis indicated that the upregulated genes in CLP-induced septic hearts were regulated mainly by NFκB1, Sp1, Jun and Egr1 (Figure 4C). Induction of sepsis through LPS administration was found to result in a higher number of upregulated genes. Notably, the top 10 upregulated genes were Saa3, Cxcl13, Acod1, Fpr1, Lcn2, Ms4a4c, Serpina3m, Marco, Ms4a8a and Chil, which are involved in various biological processes and are expressed in different cells within the mouse heart. The upregulation of these genes suggests that they play a role in the body’s response to sepsis induced by LPS administration. Upregulated genes in LPS-induced sepsis were primarily involved in innate immune response, inflammatory response and positive regulation of response to external stimulus (Figure 5A). On the other hand, the top 10 downregulated genes were Lingo3, H2-Eb1, Gck, Slc25a33, H2-Aa, Igsf1, Tuba4a, Lgi3, Aldob, and Fcrls. These downregulated genes were involved in phagosomes, peptide–ligand binding receptors and developmental biology (Figure 5B). Transcriptional network analysis showed that NFκB, Jun and Irf1 mainly regulate the expression of genes in LPS-induced sepsis (Figure 5C). These findings revealed that CLP- and LPS-induced septic hearts exhibit distinct transcriptional changes, reflecting the different molecular mechanisms and biological processes involved in the body’s response to sepsis.

### 3.3. Functional Mechanism Comparison between CLP- and LPS-Induced Sepsis

The comparative functional enrichment analysis of regulated genes in both septic hearts revealed a shared involvement primarily in the mediating inflammatory response, regulation of reactive oxygen species metabolic processes, leukocyte apoptotic processes, and positive regulation of the toll-like receptor signaling pathway (Figure 6). Moreover, we observed that specific biological functions and pathways were downregulated differently in septic hearts, depending on the induction method. For instance, CLP-induced septic hearts exhibited downregulation of the PI3K-Akt signaling pathway, calcium signaling, and HIF-1 pathways, while LPS-induced septic hearts displayed downregulation of polysaccharide metabolic processes, glycogen metabolic processes, and glucan metabolic processes.

Functional enrichment of DEGs between CLP- and LPS-induced septic hearts revealed that numerous genes associated with mitochondrial function were downregulated, such as Slc25a33, Tsfm, Mrps22, Mrpl45, Taco1, Sdhaf4, and Sco1 (Figure 7). Mitochondria play a vital role in energy production and various other cellular processes, with their dysfunction potentially leading to cellular stress and impaired heart function [32,33]. Furthermore, certain apoptotic processes can be regulated exclusively by a specific sepsis induction method (Figure 7). For example, the neutrophil apoptotic process can potentially be regulated by the CLP induction method, whereas LPS influences apoptotic processes in endothelial cells, neurons, and myeloid cells. We observed that endothelial cells respond differently to the two sepsis induction methods: CLP-induced sepsis positively regulates endothelial cell migration, while LPS-induced sepsis negatively regulates endothelial cell proliferation.

### 3.4. Gene Regulation Network Comparisons between CLP and LPS Sepsis Inductions

Our analysis revealed that both JUN and EP300 play crucial roles in the two sepsis induction models (Figure 8A,B). In particular, JUN can directly activate sepsis-related genes, altering biological mechanisms associated with essential sepsis functional genes (Figure 8A,B). Notably, Mmp3, a gene involved in the regulation of cardiac remodeling, can be directly regulated by JUN for both sepsis induction methods (Figure 8C,D). These findings highlight the significance of JUN and EP300 as central regulators in CLP- and LPS-induced septic heart gene regulation networks, potentially influencing key molecular pathways and functional outcomes in sepsis-induced cardiomyopathy. Further investigation into the specific roles and downstream targets of these transcription factors may provide valuable insights into the molecular mechanisms underlying septic cardiomyopathy as well as reveal novel therapeutic targets.

## 4. Discussion

Septic shock is a severe medical condition that arises due to a systemic inflammatory response to infection, resulting in a range of life-threatening symptoms such as hypotension, ischemia, multiple organ failure, and even death [8]. To understand the underlying molecular pathways involved in the pathogenesis of sepsis, numerous in vivo and in vitro models have been developed to explore candidate therapeutic agents aiming to mimic sepsis in humans and its treatment [34,35,36,37,38,39,40,41,42]. Although gene expression patterns during sepsis have been studied in different organs, the common set of genes induced by both methods, particularly in the heart, remains unclear.

In this study, we employed RNA sequencing to analyze changes in transcriptional profiles and molecular pathways in heart tissue samples from mice induced with sepsis through lipopolysaccharide (LPS) or cecal ligation and puncture (CLP). Our results demonstrate that LPS and CLP are effective methods for inducing septic cardiomyopathy and mortality. Compared to their respective controls, both models of sepsis lead to the activation or inhibition of different signaling pathways, resulting in altered transcription of genes. Our study specifically highlights the differential expression of matrix metalloproteinases (MMPs), such as Mmp2-3, Mmp8-9, Mmp14, Mmp17, Mmp19, and Mmp28-29, with significantly higher expression in LPS-induced sepsis compared to CLP-induced sepsis. MMPs are known to regulate cardiac remodeling, and elevated MMP expression is strongly associated with left ventricular dysfunction in heart failure patients [43,44]. Mmp9 is a commonly induced gene among the MMPs in both models of sepsis, and previous studies have reported its upregulation in the liver of CLP-induced septic rats [45]. Targeting MMPs could potentially mitigate cardiac dysfunction and improve outcomes in septic cardiomyopathy. Therapeutic approaches aimed at MMP inhibition or modulation are being explored to attenuate cardiac remodeling and dysfunction in sepsis [45]. The development of specific inhibitors targeting matrix metalloproteinases (MMPs) could serve as potential therapeutic agents for septic cardiomyopathy.

In this study, we first compared gene expression data from heart tissue samples of mice induced with sepsis through LPS or CLP with their respective control groups to identify differential gene expression. Our findings revealed that both models of sepsis induce different sets of inflammatory genes. Specifically, LPS-induced sepsis activates acute inflammation via Toll-like receptor (TLR) pathways, while CLP-induced sepsis activates inflammation via STAT3 and cytokine expression. In addition to inflammatory genes, we identified differential expression of several unique genes responsible for metabolism and mitochondrial function in both models of sepsis.

Next, we compared each group of differentially expressed genes to identify common genes regulated by each model of sepsis. We found that among 2109 differentially expressed genes, 24% were commonly regulated by both models of sepsis. Using bioinformatics analysis, we identified major differentially expressed genes and distinct gene clusters that regulate inflammation, mitochondrial respiration, oxidative stress, endothelial permeability, extracellular matrix, and apoptosis. Previous studies have shown that LPS administration increases the expression of inflammatory genes by activating NFκB and promotes cardiac tissue rearrangement [46], while RNA-sequencing of CLP-induced septic hearts revealed increased expression of Cxcr2, Cxcl1, Uchl1, Ace2, Itgb6, Lrg1, Hmgb2, Itgam, and Icam-1 genes [47]. In our study, we observed upregulation of TNF, Toll-like receptor, and NFκB signaling pathways in LPS-induced septic heart tissues, whereas CLP-induced sepsis upregulates JAK-STAT, IL-17, and B-cell receptor signaling pathways.

Although genome-wide association studies (GWAS) and high-throughput OMICs technologies have suggested several common biomarkers of sepsis, including tumor necrosis factor α (TNFα), interleukin (IL)-1, IL-6, IL-10, and macrophage migration inhibitory factor [40,41,42,48,49], the exact molecular markers associated with septic cardiomyopathy are not clearly known. Despite significant progress in medicine, the management of sepsis remains largely limited to anti-infective measures, fluid resuscitation, maintenance of multi-organ function, and other comprehensive therapies [50,51]. Several of the top regulated genes that are shared between CLP-induced and LPS-induced septic hearts, such as Cxcl13, Lcn2, and Serpina3m, have been implicated in the development of endothelial hyperpermeability and human sepsis [48,49,50,51]. In addition, this suggests that the upregulation of these genes may contribute to the mechanisms underlying septic heart failure by promoting increased dysfunction of the cardiac endothelial barrier. Therefore, the identification of commonly expressed genes in cardiac tissue across diverse models of sepsis may provide new insights into the pathogenesis of this condition and the development of novel therapies aimed at ameliorating septic cardiomyopathy.

Nonetheless, it is important to acknowledge the limitations of our study. Our analysis was limited to gene expression patterns and pathways using samples specific to mouse heart tissues. Consequently, our results may not fully capture the overall molecular mechanisms involved in sepsis-induced cardiomyopathy in humans. Moreover, the genetic and immunological differences between mice and humans could limit the direct translation of our findings to clinical settings. The heterogeneity of sepsis and the complexity of the human immune response also pose significant challenges to the applicability of our results.

In conclusion, our in-depth analysis of septic mouse models induced by LPS or CLP unraveled unique and shared gene expression patterns and pathways associated with each method of sepsis induction. The higher expression of MMPs in LPS-induced sepsis compared to CLP-induced sepsis indicates the potential role of these enzymes as therapeutic targets for septic cardiomyopathy. Our work also identified common genes involved in various processes such as inflammation, mitochondrial function, oxidative stress, endothelial permeability, extracellular matrix, and apoptosis. The identification of these shared genes enhances our understanding of the molecular mechanisms underlying septic cardiomyopathy and could pave the way for the development of novel therapies. Future studies should consider expanding the scope of research to other organs affected by sepsis and include a wider range of sepsis models. This could provide a more comprehensive understanding of the molecular underpinnings of sepsis and help identify more robust therapeutic targets.

## Figures and Tables

**Figure 1 genes-14-01366-f001:**
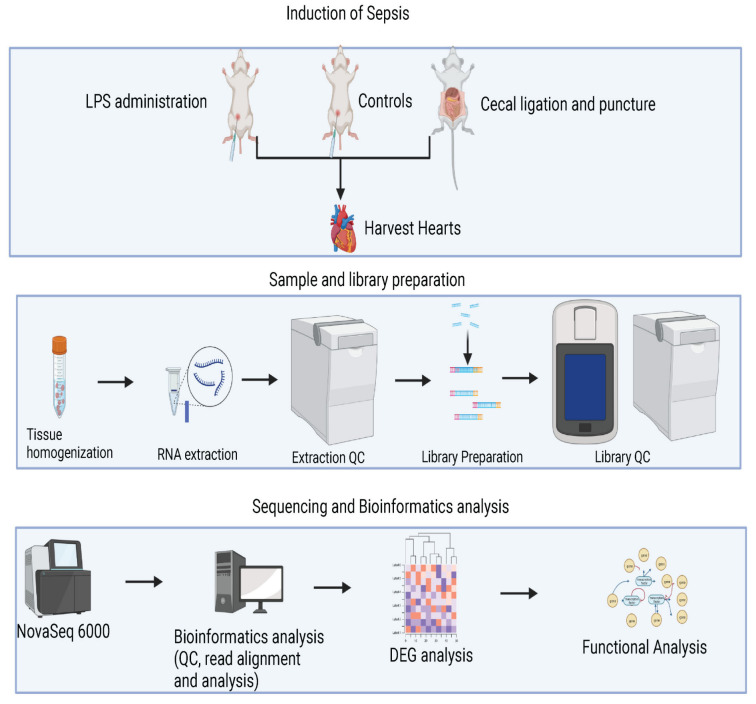
Schematic diagram of RNA-sequencing experimental design. Twenty-four hours after either LPS IP injection or CLP operation, mice were euthanized, and their hearts were rapidly excised. Sample and library preparation was followed by Illumina NovaSeq 6000 sequencing and bioinformatics analysis of DEGs. The figure was created by BioRender.com (accessed on 23 April 2023).

**Figure 2 genes-14-01366-f002:**
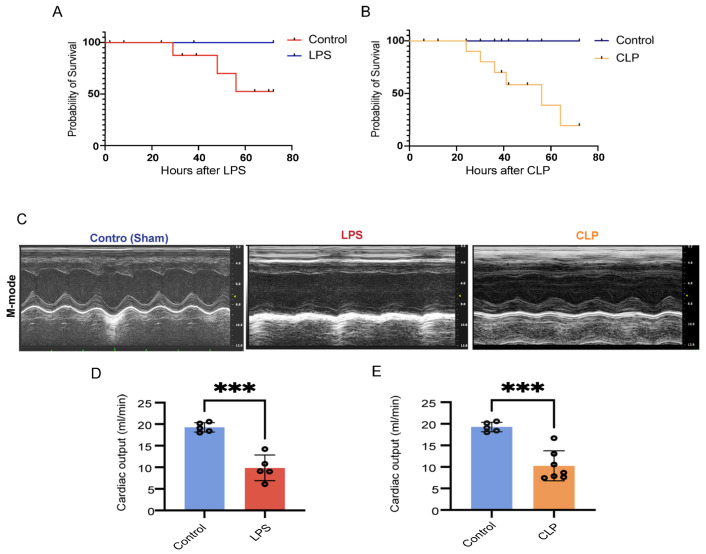
Assessment of cardiac output and mortality in septic mice. Mice weighing 20–25 g were subjected to either 40% cecal ligation and puncture (CLP, *n* = 15) or lipopolysaccharide (LPS, 5 mg/kg, *n* = 16) injection. Mortality in mice was observed over a duration of 5 days, and cardiac output was measured by echocardiography 24 h post-sepsis induction. (**A**) LPS-induced mortality (*p* < 0.001). (**B**) CLP-induced mortality (*p* < 0.001). (**C**) Representative images of M-mode echocardiography in Controls, LPS and CLP induced septic hearts. (**D**) Cardiac output in LPS-induced sepsis. (**E**) Cardiac output in CLP-induced sepsis. Data are represented as mean ± SD, *n* = 5, *** *p* < 0.001.

**Figure 3 genes-14-01366-f003:**
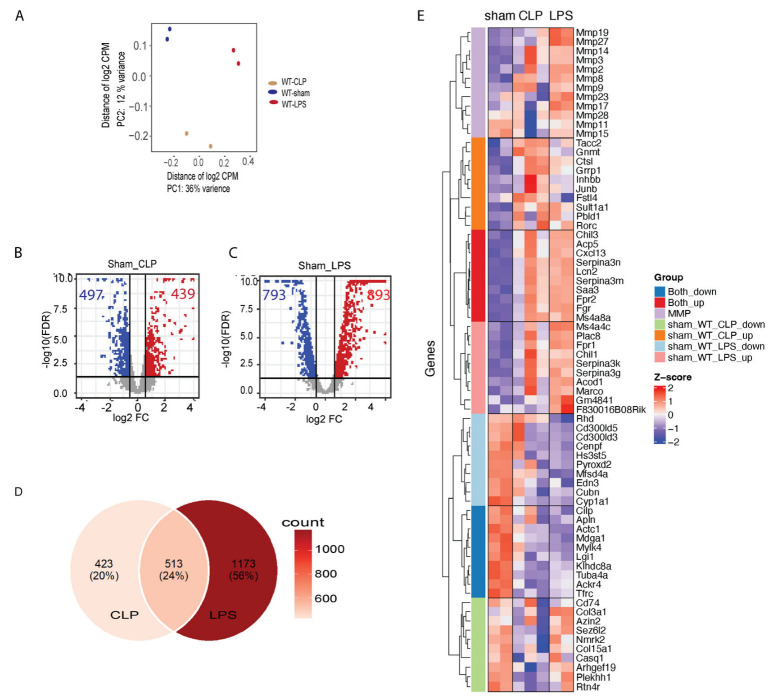
Differential gene expression and functional enrichment analysis in CLP versus LPS-induced sepsis. (**A**) Principal component analysis (PCA) plot of six samples, including duplicates for each experimental condition. Labels are as follows: Sham denotes control sham-operated samples; CLP indicates sepsis induction with cecal ligation and puncture; LPS refers to sepsis induction with lipopolysaccharide. (**B**,**C**) Volcano plot of differential gene expression (DEG) analysis, with the horizontal line indicating a false discovery rate (FDR)-corrected *p* value threshold of 0.05, and two vertical lines indicating fold change (FC) at 0.05. (**D**) Venn diagram illustrating the comparative analysis of identified DEGs from both CLP and LPS sepsis inductions. (**E**) Heatmap of selected highly expressed genes in relation to different sepsis induction methods in the heart.

**Figure 4 genes-14-01366-f004:**
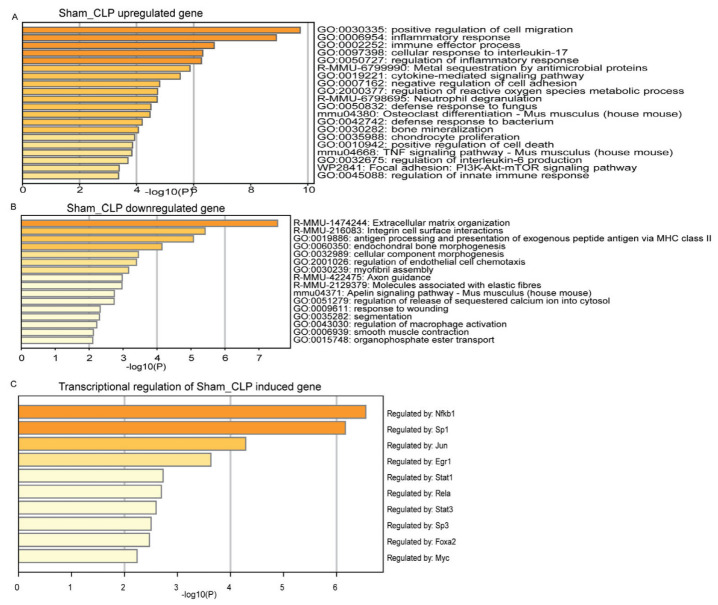
Enrichment analysis of genes in CLP-induced septic hearts. (**A**) Gene Ontology (GO) and Kyoto Encyclopedia of Genes and Genomes (KEGG) enrichment of selected upregulated genes. (**B**) GO/KEGG enrichment of selected downregulated genes. (**C**) GO-transcriptional regulatory regions in human untranslated sequences (TRRUST) enrichment of transcriptional regulators in CLP-induced samples, compared to their respective sham controls. Analyses were performed using Metascape.com (accessed on 23 April 2023).

**Figure 5 genes-14-01366-f005:**
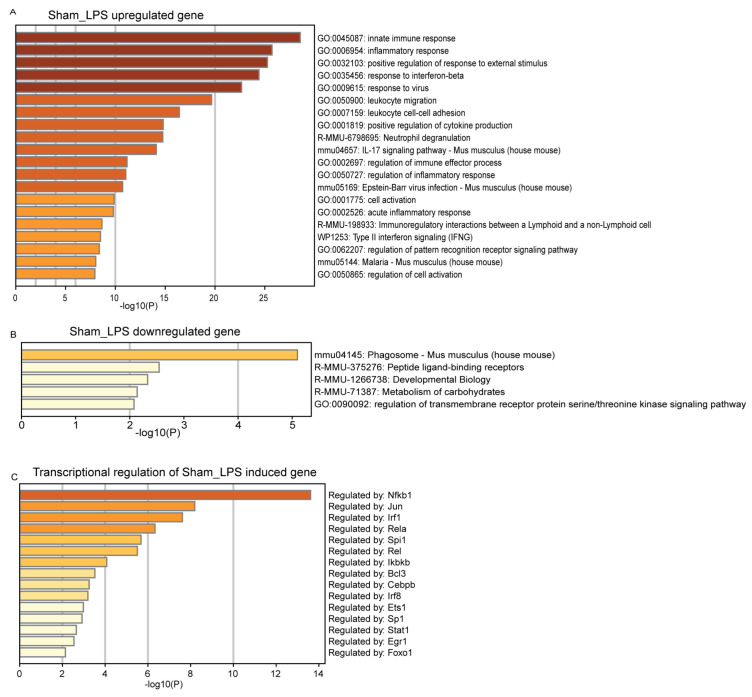
Enrichment analysis of genes in LPS-induced septic hearts. (**A**) GO/KEGG enrichment of selected upregulated genes. (**B**) GO/KEGG enrichment of selected downregulated genes. (**C**) GO-TRRUST enrichment of transcriptional regulators in LPS-induced samples, compared to their respective sham controls. Analyses were performed using Metascape.com (accessed on 23 April 2023).

**Figure 6 genes-14-01366-f006:**
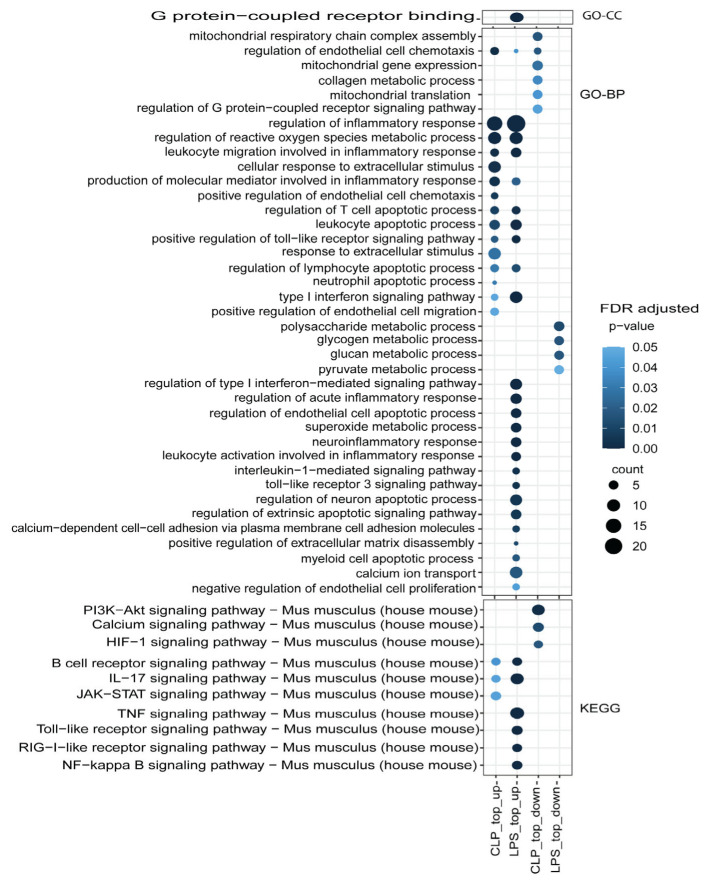
Functional enrichment analysis of differential gene expressions (DEGs). Highlighted are potentially regulated functions associated with the two sepsis induction methods, using Gene Ontology (GO) cellular components, biological processes, and KEGG databases.

**Figure 7 genes-14-01366-f007:**
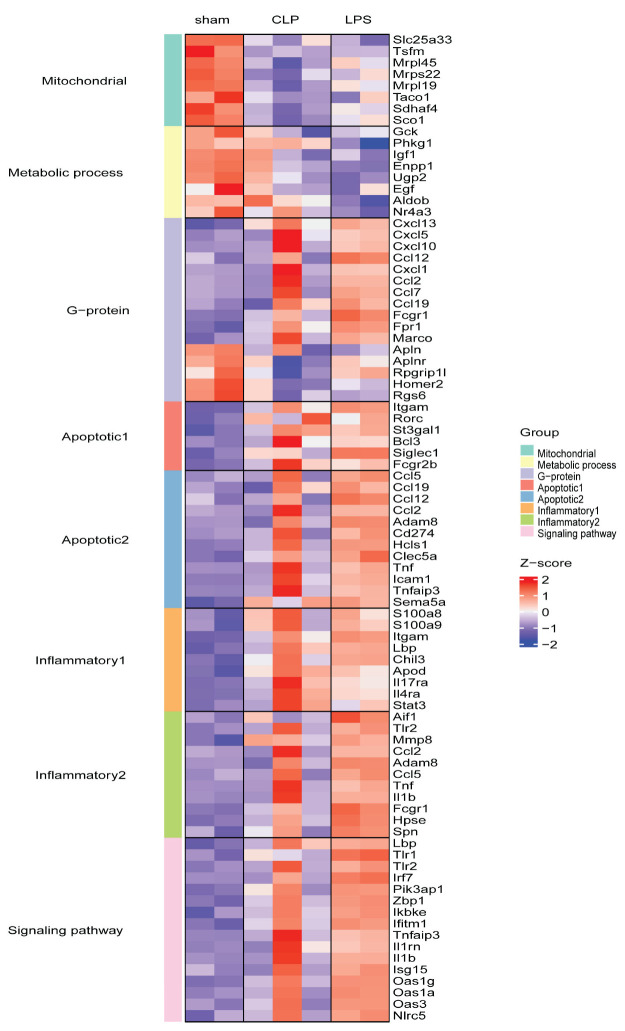
Heatmap of selected enriched functional genes. This figure represents the differential expression levels of genes.

**Figure 8 genes-14-01366-f008:**
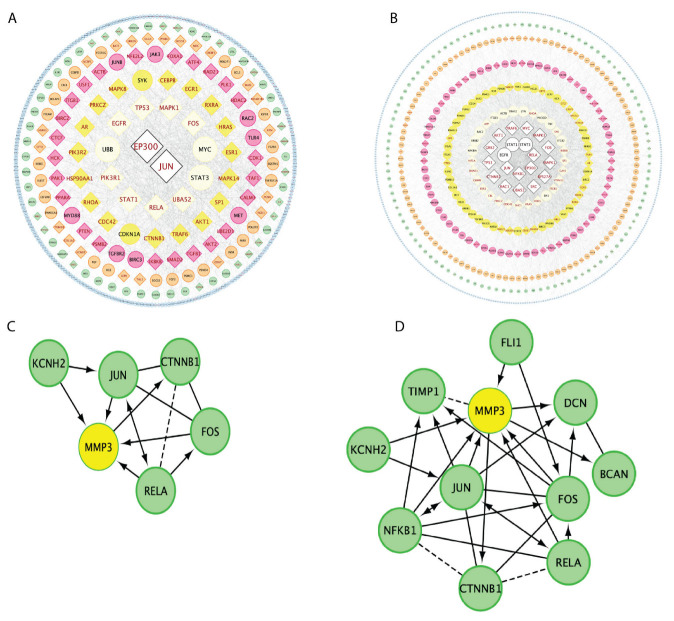
Network analysis of regulated genes. (**A**) Network of regulated genes in CLP-induced sepsis, with central, essential genes representing those with more than 20 interactions. (**B**) Network of regulated genes in LPS-induced sepsis, with central, essential genes representing those with more than 20 interactions. (**C**) Interaction network of genes involving MMP3 (Circle with yellow) in CLP-induced sepsis. (**D**) Interaction network of genes involving MMP3 (Circle with yellow) in LPS-induced sepsis.

## Data Availability

Data will be made available from the corresponding author upon request.

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
