# Peer review of "Comparative Analysis of Whole Transcriptome Profiles in Septic Cardiomyopathy: Insights from CLP- and LPS-Induced Mouse Models"

_genes, 2023, doi:10.3390/genes14071366_

Round 1
Reviewer 1 Report
In this study, the authors performed a comparative analysis of whole transcriptome profiles in two widely used mouse models of septic cardiomyopathy, the CLP model and the LPS model, to identify key genes and pathways involved in the development of septic cardiomyopathy and to evaluate the similarities and differences between the two models. In my opinion, the study is well-written and interesting. However, I have some concerns that need to be addressed, as follows:
- Abstract: This section should be rewritten to be more informative, particularly regarding the methodology used in this study and the groups of mice used.
- Introduction: The results are presented in the last paragraph of the introduction, which is not feasible. So, it is more recommended to state the objectives of this study than the results at the end of the introduction section.
- Methods: Figure 1 should be cited in the text.
- Results:
- The cardiac function should be defined as a percent change in each group and compared as a percent change among different groups from their baseline values before sepsis.
- Figure 6 includes one panel. So, there is no Figure 6A, as stated in the text.
- Please define accurately the top DEGs that are either upregulated or downregulated in each studied group and discuss them briefly in the discussion section.
- Discussion:
- Please acknowledge the limitations of this study.
- The conclusion section should be rewritten to state accurately the results of this study in a more specific manner, not in general statements, with recommendations for future studies to work on based on your study results and limitations.
- There are several grammatical, structural, capitalization, and typographical errors in the manuscript that require extensive editing by a native English speaker.
- Abbreviations should be defined in full when they first appear in the text and used thereafter.
- There are several grammatical, structural, capitalization, and typographical errors in the manuscript that require extensive editing by a native English speaker.
Author Response
Thank you for your thoughtful and constructive critique of our original paper. We greatly appreciate your feedback, and we believe that the revised version of the manuscript has been significantly improved. Below is our response to each of your comments:
Comment: Abstract: This section should be rewritten to be more informative, particularly regarding the methodology used in this study and the groups of mice used.
Response: We appreciate your input on the abstract. We have now revised this section to offer a more comprehensive and detailed summary of the study's methodology, the mouse groups used in the research, and major findings were also highlighted.
Comment: Introduction: The results are presented in the last paragraph of the introduction, which is not feasible. So, it is more recommended to state the objectives of this study than the results at the end of the introduction section.
Response: This is a very thoughtful suggestion, and we appreciate it. We have now included a clear statement of the study objectives at the end of the introduction section.
Comment: Methods: Figure 1 should be cited in the text.
Response: We apologize for the oversight. We have now appropriately cited Figure 1 within the relevant section of the methods.
Comment: Results:
The cardiac function should be defined as a percent change in each group and compared as a percent change among different groups from their baseline values before sepsis.
Response:
Thank you for your insightful comment on representing cardiac output. We understand your suggestion to present cardiac output as a percentage change from baseline. However, we believe that presenting the actual values (cardiac output is defined as the volume of blood pumped by the heart per unit of time, which is a concrete, quantifiable value of cardiac output) provides a more direct and clear understanding of the model's physiological status. Presenting absolute values allows for more straightforward comparisons across different studies, as the percentage change calculation can depend on the specific baseline conditions, which may vary. That being said, we fully appreciate the value of showing percentage changes as it can provide a normalized measure of alterations in cardiac output. To reconcile these perspectives, we have now included both the actual values in the figures and the percentage changes described in our revised manuscript. This way, the readers can appreciate the magnitude of the change in both absolute and relative terms. We hope this approach addresses your concern adequately.
Comment: Discussion:
Please acknowledge the limitations of this study. The conclusion section should be rewritten to state accurately the results of this study in a more specific manner, not in general statements, with recommendations for future studies to work on based on your study results and limitations.
Response: Thank you for highlighting the importance of acknowledging the study's limitations and providing specific conclusions. We have revised the discussion and conclusion sections accordingly to address these points.
Comment: Figure 6 includes one panel. So, there is no Figure 6A, as stated in the text.
Thank you, and it was corrected.
Please define accurately the top DEGs that are either upregulated or downregulated in each studied group and discuss them briefly in the discussion section.
Response: It has been revised accordingly. Thank you.
Comment: There are several grammatical, structural, capitalization, and typographical errors in the manuscript that require extensive editing by a native English speaker.
Response: We apologize for these errors and have enlisted the aid of a native English-speaking colleague to edit the manuscript for grammatical and typographical errors.
Comment: Abbreviations should be defined in full when they first appear in the text and used thereafter.
Response: We apologize for this oversight. We have now ensured that all abbreviations are defined in full at first mention in the text and used consistently throughout the manuscript.
Reviewer 2 Report
The authors have described a very interesting study where two differnt sepsis models have been compared. The quality of the experiments is very hogh, results are very well presented and conclusions are solid. Conclusions are bases on the experimental data.
I do have some minor suggestions.
The authiors identified difefernt autoimmune related pathways activated. It would be great to compare these experimenatl findings with some existing clincial findings, like this one PMID: 23515576. It would be interesting to see how many of these genes overlap between these two studies.
Moreover, the authors could try to separate the the cell-specific signals from the bulk RNAseq data, that might be challenging, but this article (PMID: 25545474) might be helpful to benchmark the cell-type specific profiles at least on the cytoine and inflammatory gene levels.
Author Response
Dear Reviewer, Thank you for your positive feedback on our study and for recognizing the quality of our experiments, presentation of results, and solid conclusions. We appreciate your valuable suggestions, and we have carefully considered them. Below is our response to each of your comments:
Comment 1:
The authors have identified different autoimmune-related pathways activated. It would be great to compare these experimental findings with some existing clinical findings, like this one PMID: 23515576. It would be interesting to see how many of these genes overlap between these two studies.
Response:
Thank you for your valuable suggestion. We have compared our experimental findings with existing clinical findings and identified some common shared top-regulated genes, such as Cxcl13. Interestingly, Cxcl13 has been found to be elevated in human serum sepsis and associated with cardiac microvascular permeability, suggesting a potential key role of endothelial barrier function in developing septic heart failure. Additionally, we reviewed the paper you mentioned (PMID: 23515576) titled "Peripheral blood RNA gene expression profiling in patients with bacterial meningitis." Surprisingly, none of the ten functionally relevant genes with high statistical significance (CD177, IL1R2, IL18R1, IL18RAP, OLFM4, TLR5, CPA3, FCER1A, IL5RA, and IL7R) were found in the list of shared differentially expressed genes induced by LPS and CLP in our study. Since our sequencing data is derived from heart cells rather than blood cells, the comparison with that particular study was interesting but not deemed reliable enough for inclusion in our paper. Nonetheless, we truly appreciate your thoughtful input.
Comment 2:
Moreover, the authors could try to separate the cell-specific signals from the bulk RNA-seq data, which might be challenging, but this article (PMID: 25545474) might be helpful to benchmark the cell-type-specific profiles, at least on the cytokine and inflammatory gene levels.
Response:
Thank you for commenting on the deconvolution method implementation into the bulk RNA-seq data. We are aware of the deconvolution method development and their implementations, such as CIBERSORT [1, https://www.nature.com/articles/nmeth.3337]. It has been successfully used to characterize various types of immune cells from PBMC and FL lymph node biopsies. However, the minimum sample size used in their analysis is 7. Usually, over 10 samples are used in such types of analyses. Our manuscript aims to check the "heart sepsis induction mechanisms differences," we only include 2-3 replicates in each experimental condition. Therefore, we do not have sufficient statistical power to implement deconvolution imputation to our data with appropriate deconvolution results interpretations. In general, regarding cell-specific differences, we are planning to 1) conduct single-cell (scRNA-seq) experiments for that in the future; or 2) increase the sample size for further validation of the cell-specific interpretations.
Once again, we appreciate your positive feedback and valuable suggestions. We have addressed each of your comments in our revised manuscript to enhance the clarity and relevance of our study. Thank you for your time and consideration.
Reviewer 3 Report
In this work, Ullah et al. presents a comparative analysis of the transcriptional profiles and molecular pathways in heart tissue samples from mice induced with sepsis through lipopolysaccharide (LPS) or cecal ligation and puncture (CLP) models. The topic is certainly pertinent and addresses an important aspect of septic cardiomyopathy research. The findings concerning the differential expression of Matrix Metalloproteinases (MMPs) between the two models offer important insights into septic cardiomyopathy's molecular landscape. However, there are some areas where this manuscript could be further strengthened:
1. The manuscript could benefit from a more in-depth discussion on the implications of the differential MMP expression observed between the two models. What does this mean in terms of the therapeutic approaches for septic cardiomyopathy?
2. The inclusion of additional statistical methodologies, specifically MRNET (Maximal Relevance/Minimal Redundancy Network) or GENIE3 (GEne Network Inference with Ensemble of trees), could enhance the robustness of the data analysis. These methods could be useful in exploring the complex regulatory relationships between differentially expressed genes and could provide additional depth to the presented findings.
Author Response
Dear Reviewer, Thank you for your insightful comments and your recognition of the relevance of our research topic. We have carefully considered your feedback and agree that our manuscript could be strengthened by further discussing the implications of the differential MMP expression between the LPS and CLP models. Here is our response to your comment:
Comment:
The manuscript could benefit from a more in-depth discussion on the implications of the differential MMP expression observed between the two models. What does this mean in terms of the therapeutic approaches for septic cardiomyopathy?
Response:
Thank you for your comment, and we have revised our manuscript accordingly. Targeting MMPs could potentially mitigate cardiac dysfunction and improve outcomes in septic cardiomyopathy. Therapeutic approaches aimed at MMPs inhibition or modulation are being explored to attenuate cardiac remodeling and dysfunction in sepsis. Developing specific inhibitors targeting matrix metalloproteinases (MMPs) could serve as potential therapeutic agents for septic cardiomyopathy.
Comment:
The inclusion of additional statistical methodologies, specifically MRNET (Maximal Relevance/Minimal Redundancy Network) or GENIE3 (GEne Network Inference with Ensemble of trees), could enhance the robustness of the data analysis. These methods could be useful in exploring the complex regulatory relationships between differentially expressed genes and could provide additional depth to the presented findings.
Response:
Thanks for suggesting these 2 methods used on the identified DEGs. Even though we did not use these 2 methods, we implemented other methods for network interaction and tree clustering explorations. 1). Instead of MRNET, we implemented network analysis via ReactomeFIViz in Cytoscape to compute the number of interactions with respect to each gene and grouped genes based on the number of network interactions/redundancies from high(inner) to low (outer) in Fig8A/B. 2). Instead of GENIE3, we implemented a hierarchical clustering method (‘hclust’) based on the gene expression. We updated Fig3E in our manuscript, where we can see that the same group of genes is clustered together.
We appreciate your insightful suggestions and are confident that the methodologies we implemented were suitable for our current analysis, providing valuable and robust findings.
Round 2
Reviewer 1 Report
The authors have adequately addressed all my concerns.
Minor editing of English language required